# Processed By-Products from Soy Beverage (Okara) as Sustainable Ingredients for Nile Tilapia (*O. niloticus*) Juveniles: Effects on Nutrient Utilization and Muscle Quality

**DOI:** 10.3390/ani11030590

**Published:** 2021-02-24

**Authors:** Glenise B. Voss, Vera Sousa, Paulo Rema, Manuela. E. Pintado, Luísa M. P. Valente

**Affiliations:** 1CIIMAR/CIMAR—Centro Interdisciplinar de Investigação Marinha e Ambiental, Universidade do Porto, Terminal de Cruzeiros do Porto de Leixões, Avenida General Nórton de Matos, S/N, 4450-208 Matosinhos, Portugal; gvoss@porto.ucp.pt (G.B.V.); vfsousa@icbas.up.pt (V.S.); prema@utad.pt (P.R.); 2CBQF—Laboratório Associado, Centro de Biotecnologia e Química Fina, Escola Superior de Biotecnologia, Universidade Católica Portuguesa, Rua Diogo Botelho 1327, 4169-005 Porto, Portugal; mpintado@porto.ucp.pt; 3ICBAS—Instituto de Ciências Biomédicas de Abel Salazar, Universidade do Porto, Rua de Jorge Viterbo Ferreira, 228, 4050-313 Porto, Portugal; 4UTAD—Universidade de Trás-os-Montes e Alto Douro, Quinta de Prados, 5001-801 Vila Real, Portugal

**Keywords:** bioprocess, circular economy, growth, Nile tilapia, okara digestibility, soybean by-products

## Abstract

**Simple Summary:**

The consumption of soy products increases worldwide and generates large amounts of by-products, which are often discarded. Okara is a soybean by-product with high nutritional value. This work evidenced the great potential of okara meal, after appropriate technological processing, to be used as feed ingredient in Nile tilapia diets. It was clearly demonstrated the effectiveness of the autoclave and the use of proteases from *C. cardunculus* without fermentation to increase okara nutrient bioavailability. The inclusion of up to 20% okara meal in diets for tilapia did not affect growth performance, nutrient utilization, or whole body composition compared to a reference diet. Flesh quality remained largely unaffected, although fish fed with AOK diets had the highest density of muscle fibers.

**Abstract:**

The apparent digestibility coefficients (ADCs) of differently processed okara meals were assessed in Nile tilapia diets: dried okara not autoclaved (FOK), dried okara autoclaved (AOK), okara hydrolyzed with Alcalase (ALOK) or *Cynara cardunculus* proteases (CYOK), and hydrolyzed okara fermented with lactic bacteria: *Lactobacillus rhamnosus* R11 (CYR11OK) or *Bifidobacterium animalis* ssp. *lactis* Bb12 (CYB12OK). Okara processing significantly affected nutrient digestibility: dry matter ADC was highest in CYR11OK (80%) and lowest in FOK (40%). The lowest protein digestibility was observed in CYR11OK (72%), and the highest in AOK (97%) and CYOK (91%), evidencing the effectiveness of the autoclave and the use of *C. cardunculus* proteases to increase okara protein bioavailability. The inclusion of up to 20% of AOK or CYOK did not affect fish growth, nutrient utilization, or whole body composition of Nile tilapia. The flesh quality (color, pH, water activity, cohesiveness, elasticity and resilience) was not affected by the dietary incorporation of AOK or CYOK. Fish fed with AOK diets stand out for their high density of muscle fibers, particularly in AOK20, which can explain their high muscle firmness and may result in further hypertrophic growth. Altogether, results suggest that hydrolyzed or autoclaved okara are valuable ingredients for Nile tilapia diets.

## 1. Introduction

The soybean is an important oilseed consumed worldwide, especially in Asian countries [1]. The world harvest of this oilseed in 2018 was of 348.7 million tons [2]. In general, soybean has a great commercial interest, especially due to its derived products, namely soy oil, soy proteins, soy beverage, and derivatives. Among plant protein ingredients, soybean meal has high nutritional value [3] and is the feedstuff most widely used by the aquaculture industry. However, soybean meal contains a range of antinutritional substances, such as phytic acid, protease inhibitors, and saponins [4,5], that often limits its dietary inclusion level. The high inclusion of soybean in fish diets can result in the limitation of certain essential amino acids, reduce nutrient digestibility and diet palatability [6], and can be associated with fish intestinal abnormalities [7]. On the other hand, the growing use of soy in aquafeeds not only directly competes with human nutrition, but also plays an increasing role in promoting agricultural expansion and LUC (land-use change) in the tropics [8]. As a results, in the last years, the aquaculture industry has been looking for suitable and more sustainable ingredients to include in aquafeeds [9,10,11].

The consumption of soy products increases worldwide either whole or as an ingredient in soy beverage and tofu [12]. As a consequence, the food industry generates large amounts of by-products, which are often discarded, but that can still be valuable sources of nutrients and natural antioxidants for animal nutrition [13,14,15,16]. Okara is a soybean by-product that remains after filtering the water-soluble fraction during the production of soya beverage and tofu [15]. A large amount of okara is annually produced, considering that from every 1 kg of soybean processed into soymilk/tofu, 1.1 kg of fresh okara is produced [17]. In Asian countries, the amount of okara produced per year reaches 800,000 tons in Japan, 310,000 tons in Korea, and 2,800,000 tons in China [18]. This by-product has a high moisture content (70–80%), which makes it susceptible to spoilage, being often discarded [18], but dried okara still has high protein (~30%), lipid (10–20%), high polyunsaturated fatty acids PUFA content (ca. 56% of total fatty acid content), and fiber (20–50%) levels [15,16,19]. Therefore, the processing and stabilization of fresh okara to generate a stable flour is essential, and previous studies showed that under certain conditions, its nutritional value can even be increased [19]. Moreover, and contrarily to soybean meal that has a large number of antinutritional factors [4,20], the production of soya beverage includes a thermal process that reduces most antinutritional factors [21]. Previous studies have demonstrated the effectiveness of different technological processes in the reduction of soybeans’ antinutritional factors, including autoclaving [22] and lactic fermentation. In fact, fermentation not only reduced soybeans’ antinutritional factors and indigestible carbohydrates, but also improved the digestibility of lipids in Atlantic salmon, *Salmo salar* [23]. Moreover, the fermentation of soybean meal by *Lactobacillus* was shown to enhance fish performance and nutrient digestibility; the fermented soybean also reduced the pathomorphological changes in the distal intestine in turbot [24]. Likewise, some processing techniques, including fermentation, hydrolysis, and thermal treatment, can further enhance the nutrient bioavailability of okara [25], thus improving digestibility and assimilation of nutrients by fish [26]. On the other hand, protein hydrolysates have gained prominence in recent years due to the antihypertensive, antioxidant, and antidiabetic activity of bioactive peptides [27,28,29,30].

Nile tilapia, *Oreochromis niloticus*, is one of the most common finfish species farmed worldwide [31]. It is an omnivorous fish with a dietary protein requirement estimated between 26% and 40% (as fed basis) [32]. Although fishmeal is the best protein source for tilapia and other fish due to its equilibrated amino acid profile, alternative protein sources have been widely considered [10]. Previous studies showed that the fish meal protein could be replaced by soybean meal in Nile tilapia diets without any adverse effect on growth performance, nutrient digestibility, and physiological condition [6,33,34]. Moreover, the use of soybean by-products is gaining attention due to their large worldwide production and associated environmental problems [35]. Consequently, okara was recently considered a nutritive ingredient for incorporation in fish diets, in particular for omnivorous species like Nile tilapia [36]. The dietary inclusion of okara depends on processing and stabilization techniques able to generate a stable high quality flour [19], but the nutritional value of such feedstuffs for aquafeeds remains to be evaluated.

The present study has a dual objective: (1) to assess the in vivo apparent digestibility coefficients (ADCs) of differently processed okara meals in Nile tilapia; and (2) to further evaluate the potential use of the most digestible okara meals as alternative protein sources in tilapia juveniles.

## 2. Materials and Methods

The present study was performed by accredited scientists in laboratory animal science by the Portuguese Veterinary Authority (1005/92, DGV-Portugal, following FELASA category C recommendations) and conducted according to the Directive 2010/63/EU of the European Parliament and the Council on the protection of animals for scientific purposes. The experimentation was approved by UTAD animal welfare body (382-e-DZ-2017).

### 2.1. Okara Processing and Experimental Diets

The okara generated by soy beverage producer (NUTRE Industry, Aveiro, Portugal) was used as test ingredient for Nile tilapia diets after appropriate technological processing or application of bioprocess, such as enzymatic hydrolysis and acid lactic fermentation, as shown in Table 1.

The fresh okara was divided into two lots: one was immediately dried at 65 °C until constant weight (FOK), whereas the other was firstly autoclaved, then dried under the same condition (AOK). The fresh okara was further hydrolyzed using two enzymes—Alcalase 2.4 L FG (Novozymes, Bagsvaerd, Denmark)—ALOK, or proteases from *Cynara cardunculus* (Formulab, Maia, Portugal)—CYOK. The enzymatic hydrolysis with Alcalase was performed in phosphate buffer (0.025 M), pH 8.3, E/S ratio of 0.7 (*v*/*w*) in an orbital shaker at 50 °C for 5 h. The enzymatic hydrolysis with proteases from *C. cardunculus* was performed in citric acid–sodium citrate buffer (0.1 M), pH 5.2, E/S ratio of 1.1 (*v*/*w*) in an orbital shaker at 55 °C for 2.5 h. All hydrolysates were autoclaved after the enzymatic reaction for stopping the hydrolysis and for the elimination of antinutritional factors. The okara hydrolyzed with *C. cardunculus* was further added to fructose (3% *w*/*v*) and fermented. The fermentation was performed using two lactic bacteria: *Lactobacillus rhamnosus* R11 from Lallemand (Montreal, QC, Canada) (CYR11OK) and *Bifidobacterium animalis* ssp. *lactis* Bb-12 from Christian Hansen (Hørsholm, Denmark) (CYB12OK), each fermentation was conducted at 37 °C and 120 rpm, until it reaches a pH of 4.5 (ca. 13 h). After drying, all samples were ground in a mill (Retsch, Haan, Germany) with 1 mm sieve. The proximate composition of each test ingredient is presented in Table 2.

For the digestibility trial, a commercial-based diet for Nile tilapia was formulated by SPAROS, Lda (Portugal), and used as a basal mixture. To this mixture, 1% chromic oxide (Cr_2_O_3,_ Merck KGaA, Gernsheim, Germany) was added as inert marker for the evaluation of the apparent digestibility coefficient (ADC). The reference diet (REF, Table 3) consisted of 100% of the basal mixture. Six test diets were produced by mixing 70% of the basal mixture and 30% of each test ingredient (FOKd—fresh okara meal diet; AOKd—autoclaved okara meal diet; ALOKd—enzymatic hydrolyzed (Alcalase) okara meal diet; CYOKd—enzymatic hydrolyzed (*C. cardunculus* proteases) okara meal diet; CYR11OKd—enzymatically hydrolyzed (*C. cardunculus* proteases) and fermented with R11 okara meal diet and CYB12OKd–enzymatically hydrolyzed (*C. cardunculus* proteases) and fermented with Bb12 okara meal diet). The experimental diets were further extruded by SPAROS, and pellet size was 3 mm. Upon extrusion, all batches were dried in a convection oven (OP 750-UF, LTE Scientifics, Oldham, UK) for 3 h at 60 °C and allowed to cool at room temperature.

For the growth trial, four isolipidic and isonitrogenous diets for Nile tilapia juveniles were formulated with selected okara meals either included at 10% (AOK10 and CYOK10), or 20% (AOK20 and CYOK20), resulting in a 30% and 60% replacement of plant protein sources, mainly soybean, in a lower level wheat and corn meal. Formulation and chemical composition of the experimental diets is shown in Table 4. The diets were supplemented with L-Lysine and DL-Methionine to meet the amino acid requirements for Nile tilapia [37]. Again, diets were extruded by SPAROS, and the pellet size was 3 mm, as described above.

### 2.2. Digestibility Trial

The digestibility trial was conducted in the Fish Culture Experimental Unit of UTAD (Vila Real, Portugal), with Nile tilapia (*O. niloticus*) juveniles (monosex all-male population) produced in the facility. Three homogeneous groups of 18 fish (weight 154.2 ± 1.4 g) were randomly distributed by three cylinder–conical tanks of 75 L in a recirculation freshwater system at 25 °C with automatic feces collection system (*Choubert* system) and fed the reference diet.

After the conditioning period of one week, the experimental diets were fed once a day until apparent satiation to adapt to each experimental diet before the feces collection began. Fish continued to be fed once a day during the period in which feces were collected. Approximately 30 min after feeding, each tank was carefully cleaned to assure that no remains of uneaten feed were left in the bottom of the tank. Feces were collected every morning, before feeding, and stored at −20 °C until obtaining an amount considered enough for chemical analysis (5–8 days). At the end of the trial, feces were freeze-dried prior to analysis. Since the recirculating water system used was only constituted by three tanks, this procedure had to be repeated over time, until obtaining three replicates for each test diet. In each period, a new lot of fish, with the same size-range, from the same monosex all-male initial population was used. Fish were fasted for 24 h between diets, allowing the first five days of feeding for adaption to the new diet. The remaining procedure was performed as described above.

### 2.3. Growth Trial

The growth trial was conducted in the same Fish Culture Experimental Unit of UTAD (Vila Real, Portugal), with Nile tilapia (*O. niloticus*) juveniles (monosex all-male population) produced in that facility. Fish were individually weighed (g) and measured (total length, cm), and fifteen homogeneous groups of 20 fish (mean body weight 17.3 ± 3.6 g and mean body length 9.9 ± 0.6 cm) were randomly distributed by 300 L tanks within a water recirculation system (water temperature of 24.9 ± 0.4 °C, flow rate at 5 L min ^−1^ and 12 h light/12 h dark photoperiod regime). Each diet was randomly assigned to tanks in triplicate and fed twice a day until apparent satiation. The trial lasted 10 weeks, and fish were bulk weighed once during this period (at four weeks) to monitor and register feed consumption and weight gain. Before the growth trial began, 10 fish from the initial fish stock were collected after a 48 h fasting period and euthanized by a sharp blow on the head, and then kept at −20 °C, until initial whole body composition was analyzed. At the end of the 10 week period, fish were anesthetized with MS-222 (50 mg L^−1^) and were individually weighed (g) and measured (total length, cm). Six fish per tank were sacrificed by anesthetic overdose (150 mg L^−1^ of MS222; Sigma-Aldrich Co. LLC, Bellefonte, PA, USA); liver and viscera were weighed whilst a fillet from the dorsal muscle was collected for analysis of pH, water activity, color, and texture. For histological evaluation, a one cross-sectional body slab (2 and 3 mm thick) was removed from the region immediately before the first dorsal fin, photographed together with a scale, and half of the section was immediately frozen in liquid nitrogen and kept at −20 °C until further evaluation. Five fish per tank, sacrificed by anesthetic overdose, were also collected for whole body composition analysis and frozen at −20 °C until analysis.

### 2.4. Instrumental Texture and Color Analysis of Nile Tilapia Muscle

Muscle color was measured using a CR-400 chromameter (Konica Minolta) with an aperture of 8 mm, at standard illuminant D65 using the CIE 1976 (L*: brightness; a*: redness; b*: yellowness). The apparatus was calibrated with a white plate reference standard (Minolta Co, Ltd., Osaka, Japan). Color parameters were measured by applying the colorimeter onto the raw muscle of 18 fish from each treatment. Measurements were made above the lateral line and in three points of each fillet, and mean values were considered for each fish. After flashing, L*, a*, and b* reflected light values were recorded. The hue angle and chroma were calculated from a* and b* values according to Valente et al. [38]. Hue is the relationship between redness and yellowness and is an angular measurement of color where 0° and 90° denote red and yellow hues, respectively, which is expressed as H° = tan−1 b*/a*. Chroma is expressed as C° = ((a*^2^ + b*^2^)^1/2^) and gives information about the clarity and intensity of the color.

The fillet texture was analyzed using a TA.XT plus texture analyzer, equipped with a load cell of 5 kN and a 2.0-mm-diameter probe, and controlled by Exponent v6 software (Stable Micro Systems, Godalming, UK). Texture profile analyses were obtained by double compression (constant speed and penetration depth of 1 mm s^−1^ and 4.0 mm, respectively). Measurements were performed on each raw fillet in a position corresponding to the highest thickness. Penetration depth was selected according to the maximum distance that did not induce fibers breaking, and the following measurements were determined: firmness, elasticity cohesiveness, and resilience.

### 2.5. Histological Characterization

Muscle total dorsal cross sectional (CSA) area (mm^2^) was computed after demarcating the physical limits of the whole dorsal section using the photo taken at sampling time. Transversal white muscle sections were cut at 7 µm in a cryostat CM 1950 (Leica Microsystem GmbH, Wetzlar, Germany). Sections were stained with haematoxylin-eosin (Merk, Whitehouse Station, NJ, USA), and the morphometric study was made using an interactive image analysis system (cell^B software, version 2.6, Olympus cellSens Dimension Desktop) working with a live-image captured by CCD-video camera (ColorView Soft Imaging System, Olympus) and a light microscope (BX51, Olympus, Germany). The fiber diameter and the relative number (density) of white muscle fibers per unit area (μm^2^) (NA (fibers)) were estimated according to Valente et al. [39].

### 2.6. Chemical and Physical Analysis

Fish collected from each tank for whole body composition were ground and pooled, and moisture was determined (105 °C for 24 h). Afterwards, fish were freeze-dried, ground again, and homogenized before proceeding to further analysis. The experimental diets, ingredients, and feces collected were also ground (feces were sifted) and homogenized before analysis. Proximate composition analysis was performed in duplicate and according to the Association of Official Analytical Chemists methods [40]. All samples were analyzed for dry matter (105 °C for 24 h); ash by combustion in a muffle furnace (Nabertherm L9/11/B170, Bremen, Germany; 500 °C for 5 h); crude protein (N × 6.25, Leco N analyzer, Model FP-528, Leco Corporation, St. Joseph, MO, USA); crude fat content by petroleum ether extraction using a Soxtherm Multistat/SX PC (Gerhardt, Germany); and gross energy was determined in an adiabatic bomb calorimeter (Werke C2000, IKA, Staufen, Germany). Phosphorus content was analyzed in ingredients, diets, and feces by digestion at 230 °C in a Kjeldatherm block digestion unit followed by digestion at 60 °C in a water bath and absorbance determination at 820 nm (adapted from AFNOR V 04-406). Ingredients and diet were analyzed regarding their total dietary fiber (TDF) and insoluble dietary fiber (IDF) content according to Goering and Van Soest [41]. Chromic oxide content in diets of digestibility test and feces was determined according to Bolin et al. [42]. Water activity (a_w_) was determined in muscle samples at 25 °C in an Aqualab Series 3 (Aqualab Series 3, Decagon Devices Inc., Pullmam, WA, USA), and pH was measured using a pH meter for surface equipped with a glass electrode (Crison, Barcelona, Spain); both analyses were performed in triplicate.

### 2.7. Amino Acid Analysis

Samples from test ingredients and experimental diets were hydrolyzed with HCl (6M) at 105 °C over 20 h in sealed glass vials with previous injection of nitrogen to remove the oxygen, then the pH was adjusted for 9.5 in all samples. Amino acids content for all sample were performed by derivatization with orthophthalaldehyde (OPA) methodology. The amino acids were separated by HPLC (Beckman coulter. California, CA, USA) coupled to a fluorescence detector (Waters, Milford, MA, USA). Briefly, 100 μL of each sample, at a concentration of 10 mg mL^−1^, was derivatized according to Proestos et al. [43] procedure, and injection volume of derivatives was 20 μL. The analysis was made in duplicate and quantified using a calibration curve built with amino acids pure standards (Sigma-Aldrich. St. Louis, MO, USA). Results were expressed as mg g^−1^ of sample.

### 2.8. Calculations

The apparent digestibility coefficients (ADCs) of the experimental diets were calculated according to Maynard et al. [44]: dry matter ADC (%) = 100 × (1 − (dietary Cr_2_O_3_ level/feces Cr_2_O_3_ level)) and nutrients’ ADC (%) = 100 × (1 − (dietary Cr_2_O_3_ level/feces Cr_2_O_3_ level) × (feces nutrient or energy level/dietary nutrient or energy)). The ADCs of nutrients and energy of the test ingredients were estimated according to NRC (2011): ADC_ing_ (%) = ADC_test_ + [(ADC_test_ − ADC_ref_) × ((0.7 × D_ref_)/(0.3 × D_ing_))]; where ADC_test_ = ADC (%) of the experimental diet, ADC_ref_ = ADC (%) of the reference diet, D_ref_ = % nutrient (or kJ g^−1^ gross energy) of the reference diet (Dry matter (DM) basis); D_ing_ = % nutrient (or kJ g^−1^ gross energy) of the test ingredient (DM basis).

Average body weight (ABW) = (final body weight + initial body weight)/2; Nitrogen (N), Protein (P), Lipid (L) or Energy (E) gain = (final carcass N, P, L, or E content − initial carcass N, P, L, or E content)/ABW/days.

DM, L, P, or E retention: 100 × (final body weight × final carcass nutrient content) − (initial body weight × initial carcass nutrient content) /DM, lipid, protein or energy intake; Fulton’s condition factor (K) = (final body weight/(final body length)^3^) × 100; Voluntary feed intake (VFI) = 100 × feed intake/ABW/day; Feed conversion ratio (FCR) = dry feed intake/weight gain; Protein efficiency ratio = weight gain/crude protein intake; Hepatosomatic index (HSI) = 100 × liver weight/body weight; Viscerosomatic index (VSI) = 100 × weight of viscera/body weight; Specific growth rate (SGR) = (Ln final body weight − Ln initial body weight) × 100/days.

### 2.9. Statistical Analysis

Data were tested for normality and homogeneity of variances by Shapiro–Wilk and Levene’s tests, respectively, and log-transformed whenever required before being submitted to a one-way ANOVA with the statistical program IBM SPSS STATISTICS version 23.0. When this test showed significance, individual means were compared using HSD Tukey Test. For histological analysis of % fiber, a non-parametric test (Mann–Whitney) was used instead. In all cases, significant differences were considered when *p* < 0.05.

## 3. Results

### 3.1. Digestibility Trial

The proximate composition of the test ingredients is presented in Table 2. The crude protein content of the differently processed okara meals ranged from 19% to 32%, crude fat varied between 9% and 17%, and gross energy from 20 to 23 kJ g^−1^. Phosphorus content ranged from 0.3% to 1.1% and the total fiber varied from 20% to 29% among test ingredients.

The apparent digestibility coefficients (ADCs) of the diets used in the digestibility trial and those of the test ingredients are reported in Table 5. The protein ADC was high for all diets (85 to 90%), but the CYR11OK-d showed a significantly lower value compared to the REF-d, whereas AOK-d had the highest ADC value. The CYR11OK-d and CYB12OK-d showed the highest value for dry matter ADC (75%), while FOK-d registered the lowest dry matter ADC (63%). Energy digestibility of the diets ranged between 74% to 81%; FOK-d showed the lowest energy ADC value (*p* < 0.05) and differed significantly from all other diets. Finally, phosphorus digestibility ranged from 60% to 82%, and the FOK-d also showed the lowest value, while ALOK-d presented a significantly higher value than the REF-d (*p* < 0.05).

Concerning the ADC of the ingredients tested, the dry matter ADC values varied between 40% to 80%, and FOK meal had the lowest value (*p* < 0.05). FOK showed the lowest lipid (93%) and energy (63%) ADC value. Protein ADC values differed significantly among ingredients (72–97%), and the AOK showed the highest ADC value (97%), followed by CYOK (91%). These two ingredients were hence selected to include in the experimental diets for the growth trial.

### 3.2. Growth Trial

After 10 weeks of feeding the experimental diets, all groups of fish showed a six-fold increase of their initial body weight (Table 6). Mortality was very low (<1.7%) and only observed in the reference and CYOK20 groups. All diets were well accepted, resulting in similar voluntary feed intake among dietary treatments. Final body weight (103.5 ± 31.5 g) and length (16.9 ± 2.3 cm) and the overall growth performance (SGR) of the fish remained similar among the different experimental diets. There were no significant differences among dietary treatments for the FCR (1.1) and PER (2.1–2.2) values.

The hepatosomatic and viscerosomatic indexes of fish remained similar among experimental conditions (Table 6). The dietary inclusion of different okara meals (autoclaved or hydrolyzed) did not affect significantly the final whole body composition of the fish. Although lipid retention was highest in fish fed the REF diet, nutrient gain remained similar among dietary treatments (Table 6).

### 3.3. Flesh Quality Traits

The instrumental color and texture properties of Nile tilapia muscle were evaluated by instrumental analysis (Table 7), and no significant differences were observed in fillet color among dietary treatments by the end of the growth trial. The texture properties, cohesiveness, elasticity, and resilience of the fillets (Table 7) were not significantly affected by the experimental diets. However, the muscle firmness varied significantly; the firmness of fish fed the experimental diets was generally higher than that of fish fed the reference diet, but differences were only significant in AOK20. The muscle a_w_ and pH values did not differ among dietary treatments.

Muscle cellularity results are presented in Table 8. The total muscle cross-sectional area (CSA) and total number of fibers did not vary significantly (*p* > 0.05) among experimental diets. However, the diets CYOK10 and CYOK20 showed significantly larger fiber diameters than those of the REF and AOK10 diets (Figure 1).

This resulted in a significantly lower fiber density (N/mm^2^) in CYOK20 compared to diets AOK10 and AOK20. The percentage of small-sized fibers (≤30 μm) was highest in fish fed AOK diets, and this fiber class was even inexistent in fish fed CYOK20; contrarily, the highest % of large-sized fibers (>100 μm) was observed in fish fed either the REF diet or okara hydrolyzed with *C. cardunculus* extract (CYOK10 and CYOK20), but differences were not statistically significant.

## 4. Discussion

### 4.1. Digestibility Trial

The food industry generates large amounts of okara, which is often discarded but can still be a valuable nutrient source to include in fish diets. According to Mo et al. [45], this is a low cost by-product that can be used to feed herbivorous and omnivorous fish species like carp and tilapia. The present study showed that the proximate composition of okara meal varied according to the technological processing used. The addition of fructose during okara fermentation resulted in higher carbohydrates content in fermented okara meals (CYR11OK and CYB12OK). On the other hand, the use of a buffer solution during the enzymatic hydrolyses resulted in increased ash content in ALOK, CYOK, CYR11OK, and CYB12OK meals. These variations led to lower protein and fat contents in fermented okara meals (ALOK, CYOK, CYR11OK and CYB12OK). Additionally, the thermal treatment used for extraction of soya beverage eliminates most antinutrients, such as protease inhibitors and phytates, which is an advantage for fish feeding [4,19]. The first step toward the evaluation of the nutritional value of an ingredient for a certain fish species is the determination of its apparent digestibility coefficient (ADC). The present study shows that okara meal nutrients can be highly digestible by Nile tilapia, when okara is properly processed, and can reach similar values to those reported for commonly used feedstuffs [46]. In fact, the dry matter ADC values observed for the hydrolyzed and fermented okara meal ingredients (ALOK, CYOK, CYR11OK, CYB12OK) were well within the range of values previously reported for soybean ingredients (75–78%), in Nile tilapia; but in FOK and AOK, dry matter ADCs values were lower, corresponding to values reported in pea seed meal (46%) and faba bean meal (67%), respectively [46]. Moreover, Vidal et al. [47] obtained higher values for dry matter ADC in soybean coproducts (full-fat soybean meal, expeller pressed soybean meal, low-protein soybean meal, high-protein soybean meal, and soybean protein concentrate) fed to Nile tilapia compared to those presently obtained. Contrarily, in fermented soybean, Dong et al. [48] found an ADC value for dry matter (70%) lower than that observed in our study (79–80%), irrespectively of the enzymes presently used in fermented okara (CYR11OK and CYB12OK). The DM ADC value represents the overall digestibility for all components of an ingredient, and reflects the digestible fraction of both organic and inorganic matter, being largely dependent on ingredients insoluble carbohydrates and mineral composition. In the present study, it was clearly observed that applied processing technologies had a profound impact on ingredients’ chemical composition generally improving okara DM ADC. In fact, the highest vales were observed after fermentation, which might be explained by the lowest insoluble dietary fiber in those ingredients.

The ADCs for protein (72–97%) varied significantly among the test ingredients. The highest protein ADC value was obtained in okara meal submitted to thermal treatment (AOK) (97%), demonstrating the effectiveness of the autoclave for increasing nutrient bioavailability, and possibly eliminating most antinutrients (trypsin inhibitors). Moreover, the hydrolysis performed by the two enzymes (Alcalase and *C. cardunculus* proteases) resulted in distinct protein ADC values in Nile tilapia, with CYOK exhibiting higher digestibility values than ALOK. This can be partially related to the peptide profile and free amino acids obtained after each enzymatic hydrolysis. According to Voss et al. [30], the okara hydrolyzed by Alcalase (ALOK) had a higher free amino acid content and peptide fractions with lower molecular weight than okara hydrolyzed by proteases from *C. cardunculus* that resulted in peptides with both high and low molecular weights. Extensively hydrolyzed proteins and consequently higher level of free amino acids may result in a saturation of intestinal transporters, resulting in low digestion and absorption of protein [49]. There are no previous studies reporting Okara digestibility values for direct comparison, but CYOK protein ADC corresponds well with previous observations for soybean meal [46,50,51], suggesting it can be a good protein source for Nile tilapia. Moreover, the ADC of protein in CYR11OK (72%) was much lower than values previously reported for either soybean meal or fermented soybean [46,47,48,50]. Although CYB12OK has a better protein ADC compared to CYR11OK, fermentation tended to decreased protein bioavailability in relation to CYOK.

In general, the energy ADC values depend on the energy ADCs for carbohydrate, lipid, and protein, as these constituents account for most of the dry matter [32]. In this study, energy ADC values varied significantly among okara ingredients, being lowest in fresh okara (FAOK). The lowest dry matter ADC of FAOK was reflected in a poor energy ADC value, which corresponds well with results from Ngo et al. [52] showing that the dry matter and energy ADC values are correlated. FAOK had the highest protein and fat content, and also had high levels of fiber. However, the lower energy ADC found in this ingredient may be related to some remaining antinutritional factors compared to AOK. The application of thermal treatments is generally associated with a higher digestibility of plant ingredients, including soybeans [4]. The energy ADC values in FAOK and AOK were lower than those previously reported for soybean coproducts fed to Nile tilapia [47]. However, energy ADC of hydrolyzed and fermented okara (ALOK, CYOK, CYR11OK, and CYB12OK) were similar to those reported for defatted soybean meal and full-fat toasted soybean [46,50] and even higher than those observed by Dong et al. [48] in fermented soybean and soybean meal. This might be related to the reduction of indigestible carbohydrates and antinutritional factors (reduced trypsin inhibitor activity) as previously reported in Atlantic salmon fed fermented soybean [23].

In conclusion, the present results show that nutrient bioavailability in okara ingredient can be largely improved upon proper processing, confirming previous studies with soybean coproducts [23,47]. The highest protein digestibility was observed with AOK followed by CYOK, clearly demonstrating the effectiveness of the autoclave for eliminating okara antinutrients and the use of proteases from *C. cardunculus* without fermentation for promoting peptide absorption in the intestine. These two okara meals were hence selected for a growth trial in Nile tilapia.

### 4.2. Growth Trial

Soybean meal has high nutritional value and is the most widely used feedstuff by the aquaculture industry for replacing fish meal. It is generally known that the nutritional value and amino acid (AA) profile of soybean products depends on their origin and industrial processing [19,47]. In general terms, the AA profiles of the selected okara meals selected for this study are consistent with those reported in the literature for other soybean co-products; the two most limiting amino acids, lysine and methionine levels, ranged between 30–34 and 9–11 mg AA/g of sample, in CYOK and AOK, respectively, whilst previous studies reported values of 32–53 and 5.4–8.7 mg AA/g of sample, respectively [47].

Although several studies have focused on different products from soybean to partial or totally replace fish meal in aquafeeds [6,53,54,55,56], studies using okara in fish diets are very scarce. In the present study, it was clearly demonstrated that okara meal can be included at 20% in diets for Nile tilapia, representing a 60% replacement of plant protein sources (mainly soybean meal), without affecting feed intake or nutrient utilization, whilst assuring high growth rates (2.4–2.6%). Likewise, in a previous study, El-Saidy [36] reported that the dietary inclusion of okara up to 40%, replacing 75% fish meal, did not affect the growth performance of Nile tilapia. According to Kim and Kaushik [57], similar growth should be expected in fish fed diets containing the same level of digestible energy intake and digestible protein intake. Although diets’ digestibility were not determined in the present study, the protein and energy ADC values of the selected okara ingredients (AOK and CYOK) did not differ significantly. Moreover, feed intake remained similar among treatments, evidencing the good palatability of all diets. Mamauag et al. [56] reported that the inclusion of 20% soy peptide hydrolysate in diets for Japanese flounder, *Paralichthys olivaceus*, was an effective feed attractant and improved diets palatability, due the presence of small molecular weight compounds as free amino acid. Such diets promoted growth performance and improved blood biochemical parameters. However, these results could not be confirmed in the present study with CYOK diets.

The feed conversion efficient ratio (FCR) obtained for all diets was generally lower (1.1) than values reported in previous studies using increasing levels of dried okara meal (1.7–1.8) [36] or yeast fermented soybean meal (1.6) [6] in Nile tilapia. The okara meal used in the present work was previously autoclaved, which can almost completely eliminate the antinutritional factors, mainly protease inhibitors, and thus can improve the absorption of nutrients on fish. Additionally, all experimental diets were extruded, which can further explain the best feed utilization. Previous studies employing heat treatment in soybeans have also reported improved feed utilization in channel catfish [55] and Coho Salmon [58]. The hepatosomatic and viscerosomatic indexes of fish were also similar among experimental conditions. Similar results were reported in rockfish fed different levels of soybean fermented by *Bacillus subtilis* [59] or in Nile tilapia fed increasing levels of soy peptide hydrolysate [56].

Whole body composition (dry matter, crude protein, and crude lipid) of the fish did not vary significantly between AOK and CYOK dietary treatments. Likewise, increasing dietary levels of soy peptide hydrolysate in Japanese flounder [56] or fermented soybean meal in *Myxocyprinus asiaticus* [5] for replacement of fish meal up to 55–60% did not affect whole body composition. Contrarily, dried okara meal [36] or fermented soybean [59] induced lower lipid content in rockfish and in Nile tilapia, compared to a fish-meal based control diet, whilst whole body protein levels remained unchanged.

The muscle color and textural properties of muscle are important quality attributes for consumers. Color evaluation using the tristimulus L*, a*, and b* in Nile tilapia muscle fed diets using AOK and CYOK did not vary significantly compared with the reference diet and correspond to previous values reported in literature [60]. Similar water activity and textural parameters including cohesiveness, elasticity, and resilience were observed among treatments, contrarily to firmness; fish fed OKA20 had firmer muscle than the reference diet. The muscle cellularity is directly related to growth and is one of the main determinants of muscle texture [61]. Muscle growth was not affected by the experimental diets, but a significant increase in muscle fiber diameter was observed in fish fed CYOK compared to those fed the REF or AOK10 diets. CYOK fish also showed the lowest % of small-sized fibers (≤30 μm), suggesting that an enzymatic hydrolysis of okara using the *C. cardunculus* proteases promotes muscle hypertrophic growth. In contrast, fish fed with AOK diets stand out for their high density of muscle fibers, particularly in AOK20, which can explain its high muscle firmness. Firmness was negatively correlated with the number of large-sized fibers. A negative correlation between muscle firmness and white fiber diameter was also reported in gilthead seabream from distinct production systems [39]. Moreover, the high percentage of fibers ≤ 30 μm in fish fed AOK suggests a hyperplastic growth, and may have a greater potential to further growth by hypertrophy.

## 5. Conclusions

The results obtained in this study showed that all different okara meals were well digested by Nile tilapia. The highest protein digestibility was observed in AOK followed by CYOK, demonstrating the effectiveness of the autoclave and the use of proteases from *C. cardunculus* without fermentation to increase nutrient bioavailability. The inclusion of up to 20% of AOK or CYOK did not affect fish growth, nutrient utilization, or whole body composition. Fish fed with AOK diets stand out for their high density of muscle fibers, particularly in AOK20, which can explain their high muscle firmness and may result in further hypertrophic growth. Since okara can be obtained from local food industries, its use could decrease the importation of feedstuffs and consequently decrease the carbon footprint in the aquafeed sector.

## Figures and Tables

**Figure 1 animals-11-00590-f001:**
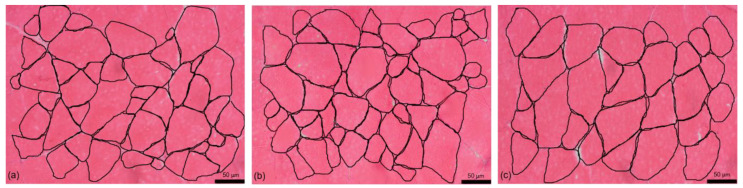
Cross sections of skeletal muscle from Nile Tilapia fed with the different experimental diets (**a**) REF; (**b**) OKA10; (**c**) CYOK20. Notice the larger fiber size and lower fiber density in CYOK20.

**Table 1 animals-11-00590-t001:** Process applied in by-product from soy beverage to obtain okara meal used as test ingredients.

Ingredient	Processing Method
Dried okara (FOK)	Fresh okara dried until constant weight (65 °C for 68 h)
Autoclaved and dried okara (AOK)	Autoclaved okara (1 atm, 121 °C for 20 min) followed by drying until constant weight (65 °C for 68 h)
Hydrolyzed Okara using Alcalase (ALOK)	Okara hydrolyzed with Alcalase and autoclaved, followed by drying until constant weight (65 °C for 68 h)
Hydrolyzed Okara using proteases from *Cynara cardunculus* (CYOK)	Okara hydrolyzed with proteases from *Cynara cardunculus* and autoclaved followed by drying until constant weight (65 °C for 68 h)
Hydrolyzed Okara using proteases from *Cynara cardunculus* and fermented with R11 ^a^ (CYR11OK)	Okara hydrolyzed with proteases from *Cynara cardunculus*, autoclaved, further added fructose and fermented with *Lactobacillus rhamnosus* R11, followed drying until constant weight (65 °C for 68 h)
Hydrolyzed Okara using proteases from *Cynara cardunculus* and fermented with Bb12 ^b^ (CYB12OK)	Okara hydrolyzed with proteases from *Cynara cardunculus*, autoclaved, further added fructose and fermented with *Bifidobacterium animalis* ssp. *lactis* Bb12 followed by drying until constant weight (65 °C for 68 h)

^a^*Lactobacillus rhamnosus* R11; ^b^
*Bifidobacterium animalis* ssp. *lactis* Bb12.

**Table 2 animals-11-00590-t002:** Proximate composition (% Dry matter, DM or kJ g^−1^ DM) of the test ingredients used in digestibility trial.

Composition	FOK	AOK	ALOK	CYOK	CYR11OK	CYB12OK
Dry matter	97.2	97.4	92.0	94.0	88.7	88.7
Ash	4.0	3.9	7.3	12.0	11.1	10.7
Crude protein	31.6	31.0	26.9	25.2	20.1	19.4
Crude fat	17.2	16.4	15.4	13.2	9.7	9.3
Gross energy (kJ g^−1^)	22.6	22.5	21.5	20.0	19.7	20.0
Phosphorus	0.4	0.4	1.1	0.3	0.3	0.3
TDF ^a^	27.0	21.8	27.3	26.2	28.6	20.3
IDF ^b^	14.7	13.6	15.2	15.0	9.6	10.6
Carbohydrates ^c^	47.2	48.7	50.4	49.6	59.1	60.6

^a^ TDF: Total dietary fiber; ^b^ IDF: Insoluble dietary fiber carbohydrates was calculated by difference. ^c^ Total Carbohydrates = 100 − Ash − Crude Protein − Crude Fat.

**Table 3 animals-11-00590-t003:** Formulation (%) and proximate composition (% Dry matter, DM or kJ g^−1^ DM) of the basal mixture and experimental diets for the digestibility trial.

Experimental Diets							
	REF	FOKd	AOKd	ALOKd	CYOKd	CYR11OKd	CYB12OKd
Basal mix	100	70	70	70	70	70	70
FOK		30					
AOK			30				
CYB12OK				30			
CYR11OK					30		
CYOK						30	
ALOK							30
**Proximate Composition**							
Dry matter	93.9	93.8	93.0	94.5	92.5	94.1	93.7
Ash	9.0	7.2	7.2	8.5	9.3	9.2	9.0
Crude protein	43.0	36.6	38.4	35.4	34.9	35.4	37.1
Crude fat	8.1	11.7	11.2	11.0	10.3	9.3	9.2
Gross energy (kJ g^−1^)	20.5	21.0	21.1	20.4	20.2	19.9	20.1
Phosphorus	1.2	0.9	0.9	1.2	0.9	0.9	0.9
TDF	22.7	25.6	22.4	15.9	22.7	23.9	23.8
IDF	8.2	10.2	10.1	8.4	11.0	12.7	7.3

The abbreviations for the experimental diets stand for: REF—reference diet; FOKd—okara meal diet; AOKd—autoclaved okara meal diet; ALOKd—enzymatic hydrolyzed (Alcalase) okara meal diet; CYOKd—enzymatic hydrolyzed (*Cynara cardunculus* proteases) okara meal diet; CYR11OKd—enzymatically hydrolyzed (*Cynara cardunculus* proteases) and fermented with R11 okara meal diet, and CYB12OKd—enzymatically hydrolyzed (*Cynara cardunculus* proteases) and fermented with Bb12 okara meal diet.

**Table 4 animals-11-00590-t004:** Formulation and proximate composition of the experimental diets and ingredient used in the growth trial.

Ingredients (%)	REF	AOK10	AOK20	CYOK10	CYOK20
Fishmeal 60 ^a^	10.0	10.0	10.0	10.0	10.0
Corn gluten ^b^	17.0	17.0	17.0	17.0	17.0
Soybean meal 48 ^c^	17.5	12.5	7.5	14.0	10.5
Rapeseed meal ^d^	15.0	15.0	15.0	15.0	15.0
Wheat meal ^e^	5.0	4.5	3.0	4.0	1.0
Wheat bran ^f^	10.0	10.0	10.0	10.0	10.0
Rice bran full fat ^g^	12.0	12.0	12.0	12.0	12.0
Corn meal ^h^	8.0	4.3	1.5	3.3	0.5
Soybean oil ^i^	2.7	2.0	1.2	2.0	1.2
Vitamin & Mineral Premix ^j^	1.0	1.0	1.0	1.0	1.0
Betaine HCl ^k^	0.2	0.2	0.2	0.2	0.2
Antioxidant ^l^	0.2	0.2	0.2	0.2	0.2
Sodium propionate ^m^	0.1	0.1	0.1	0.1	0.1
MCP ^n^	0.2	0.2	0.2	0.2	0.2
L-Lysine ^o^	0.7	0.7	0.7	0.7	0.7
DL-Methionine ^p^	0.4	0.4	0.4	0.4	0.4
AOK ^q^	-	10.0	20.0	-	-
CYOK ^r^	-	-	-	10.0	20.0
**Proximate composition (% DM or kJ g^−1^ DM)**					
Dry matter	90.5	89.9	90.9	92.3	90.1
Ash	7.2	7.1	7.1	7.8	8.4
Crude protein	42.9	43.8	43.8	43.0	43.8
Crude fat	7.2	8.3	8.8	8.0	8.3
Gross energy (kJ g^−1^ DM)	21.3	21.7	21.9	21.5	21.5
TDF ^s^	17.3	16.9	16.8	15.6	17.1
IDF ^t^	9.6	9.1	10.4	9.6	10.8
Phosphorus	2.6	2.7	2.0	2.4	2.5
**Essencial Amino Acids (mg g^−1^ Sample)**					
Arginine	20.8	20.2	22.3	21.7	23.1
Histidine	7.9	6.0	7.3	9.8	8.4
Lysine	31.8	30.9	34.2	31.7	29.7
Threonine	15.7	14.7	16.0	16.5	17.4
Isoleucine	17.8	17.3	19.1	18.5	18.9
Leucine	37.1	36.0	40.1	38.5	38.1
Valine	21.5	20.9	23.2	22.0	22.3
Methionine	9.3	9.8	10.6	10.6	9.4
Phenylalanine	17.7	17.2	19.2	18.6	17.1
Cystine	5.7	5.0	5.8	5.5	5.4
Tyrosine	13.7	13.5	15.0	14.5	14.8
Aspartic acid	53.4	52.9	58.4	57.2	59.8
Glut acid + Glutamine	99.4	96.1	105.7	104.3	109.3
Alanine	24.5	23.8	26.1	25.6	25.4
Glycine	7.4	6.3	6.4	6.8	7.8
Serine	17.7	17.2	19.2	18.8	19.8

The abbreviations for the experimental diets stand for: REF—reference diet; AOK10 and AOK20—diets with 10 and 20% autoclaved okara meal, respectively; CYOK10 and CYOK20—diets with 10 and 20% of okara hydrolyzed by proteases from extract *Cynara cardunculus* meal, respectively. ^a^ CONRESA 60: 61.2% crude protein (CP), 8.4% crude fat (CF), Conserveros Reunidos S.A., Spain. ^b^ Corn gluten meal: 61% CP, 6% CF, COPAM, Portugal. ^c^ Dehulled solvent extracted soybean meal: 47% CP, 2.6% CF, CARGILL, Spain. ^d^ Defatted rapeseed meal: 34% CP, 2% CF, Premix Lda, Portugal. ^e^ Wheat meal: 10.2% CP, 1.2% CF, Casa Lanchinha, Portugal. ^f^ Wheat brean: 15% CP, 4% CF, Ribeiro & Sousa Cereais, Portugal. ^g^ Rice bran full-fat: 13.3% CP, 16.3% CF, Casa Lanchinha, Portugal. ^h^ Corn meal: 8.1% CP, 3.7% CF, Ribeiro & Sousa Cereais, Portugal. ^i^ Soybean oil: Henry Lamotte Oils GmbH, Germany. ^j^ Vitamin and mineral premix: PREMIX Lda, Portugal. Vitamins (mg or IU.Kg^−1^ diet): DL-alpha tocopherol acetate, 100 mg; sodium menadione bisulphate, 25 mg; retinyl acetate, 20,000 IU; DL-cholecalciferol, 2000 IU; thiamin, 30 mg; riboflavin, 30 mg; pyridoxine, 20 mg; cyanocobalamin, 0.1 mg; nicotinic acid, 200 mg; folic acid, 15 mg; ascorbic acid, 500 mg; inositol, 500 mg; biotin, 3 mg; calcium panthotenate, 100 mg; choline chloride, 1000 mg, betaine, 500 mg. Minerals (mg or mg.kg^−1^ diet): copper sulphate, 9 mg; ferric sulphate, 6 mg; potassium iodide, 0.5 mg; manganese oxide, 9.6 mg; sodium selenite, 0.01 mg; zinc sulphate, 7.5 mg; sodium chloride, 400 mg; excipient wheat middlings. ^k^ Betaine HCl: Premix Lda, Portugal. ^l^ Antioxidant: Paramega PX, Kemin Europe NV, Belgium. ^m^ Sodium propionate: Premix Lda, Portugal. ^n^ Monocalcium phosphate (MCP): 22% P, 18% Ca, Fosfitalia, Italy. ^o^ L-Lysine: Biolys 54.6%, EVONIK Nutrition & Care GmbH, Germany. ^p^ DL-Methionine for aquaculture 99%, EVONIK Nutrition & Care GmbH, Germany. ^q^ AOK: Autoclaved okara meal (dry matter (95.3%), ash (4.8%) protein (38.4%), crude fat (18.2%), gross energy (23.5 kJ g^−1^), phosphorus (1.3%), TDF (20.6%), IDF (16.5%)). ^r^ CYOK: Okara hydrolyzed by proteases from extract *Cynara cardunculus* meal (dry matter (91.5%), ash (10.6%) protein (33.7%), crude fat (15.5%), gross energy (17.2 kJ g^−1^), phosphorus (1.1%), TDF (13.0%), IDF (8.4%)). ^s^ TDF: Total dietary fiber. ^t^ IDF: Insoluble dietary fiber. - stands for not determined.

**Table 5 animals-11-00590-t005:** Apparent digestibility coefficients (ADCs) of diets and test ingredients.

ADC Diets (%)	REFd	FOKd	AOKd	ALOKd	CYOKd	CYR11OKd	CYB12OKd
Dry matter	73.4 ± 0.4 ^ab^	63.2 ± 1.2 ^c^	70.7 ± 0.7 ^b^	72.9 ± 1.7 ^ab^	73.4 ± 1.0 ^ab^	75.1 ± 0.8 ^a^	74.9 ± 0.7 ^a^
Protein	87.7 ± 1.1 ^bc^	86.3 ± 0.4 ^bd^	89.8 ± 0.6 ^a^	86.3 ± 0.4 ^bcd^	88.3 ± 0.4^ac^	85.1 ± 0.8 ^d^	87.1 ± 1.0 ^bcd^
Energy	79.8 ± 0.1 ^a^	74.4 ± 1.1 ^b^	78.4 ± 1.3 ^a^	79.3 ± 0.9 ^a^	80.4 ± 0.2 ^a^	80.3 ± 1.1 ^a^	80.8 ± 0.9 ^a^
Lipids	91.8 ± 0.4 ^b^	93.1 ± 0.0 ^bc^	96.8 ± 0.7 ^a^	96.2 ± 0.2 ^a^	96.6 ± 0.2 ^a^	95.4 ± 0.9 ^ab^	96.4 ± 0.5 ^a^
Phosphorus	71.9 ± 2.0 ^bc^	60.0 ± 1.3 ^d^	73.9 ± 5.3 ^bc^	82.0 ± 2.3 ^a^	69.9 ± 1.0 ^b^	77.9 ± 1.1 ^ac^	77.5 ± 1.0 ^ac^
**ADC Ingredients (%)**		**FOK**	**AOK**	**ALOK**	**CYOK**	**CYR11OK**	**CYB12OK**
Dry matter		40.4 ± 4.0 ^c^	64.6 ± 2.2 ^b^	72.0 ± 5.8 ^ab^	73.6 ± 3.4 ^ab^	79.6 ± 2.9 ^a^	78.6 ± 2.4 ^a^
Protein		81.9 ± 1.7 ^bc^	96.6 ± 2.6 ^a^	81.5 ± 1.9 ^bc^	91.1 ± 1.8 ^ab^	72.1 ± 4.9 ^c^	84.1 ± 5.9 ^b^
Energy		63.1 ± 2.4 ^b^	75.5 ± 2.9 ^a^	78.4 ± 8.5 ^a^	81.9 ± 2.2 ^a^	81.7 ± 3.6 ^a^	83.5 ± 2.2 ^a^

Values are presented as mean ± standard deviation (*n* = 3). Values in the same row without a common superscript letter differ significantly (*p* < 0.05).

**Table 6 animals-11-00590-t006:** Final growth performance, somatic indexes, and whole body composition (% or kJ g ^−1^ of wet weight (WW)) of Nile tilapia fed the experimental diets for 10 weeks.

Variable	REF	AOK10	AOK20	CYOK10	CYOK20
**Growth**					
Initial body weight (g)	17.4 ± 3.6	17.2 ± 3.6	17.4 ± 3.8	17.3 ± 3.7	17.2 ± 3.6
Final body weight (g)	110.6 ± 35.7	103.5 ± 32.2	99.4 ± 33.6	105.6 ± 31.4	102.1 ± 26.7
Final body length (cm)	17.5 ± 2.0	17.2 ± 2.0	16.9 ± 2.1	17.2 ± 2.1	17.2 ± 1.5
^a^ K	2.22 ± 0.07	2.04 ± 0.07	2.19 ± 0.19	2.12 ± 0.05	2.12 ± 0.21
^b^ SGR	2.57 ± 0.09	2.49 ± 0.03	2.42 ±0.08	2.53 ± 0.06	2.47 ± 0.07
Voluntary feed intake (g Dry Matter/100 g ^c^ ABW/day)	2.14 ± 0.11	2.12 ± 0.05	2.13 ± 0.01	2.11 ± 0.09	2.20 ± 005
^d^ FCR	1.06 ± 0.07	1.07 ± 0.03	1.09 ± 0.03	1.05 ± 0.06	1.11 ± 0.0
^e^ PER	2.2 ± 0.1	2.1 ± 0.1	2.1 ± 0.1	2.2 ± 0.1	2.1 ± 0.1
^f^ HSI	1.3 ± 0.2	1.4 ± 0.1	1.5 ± 0.1	1.2 ± 0.3	1.2 ± 0.2
^g^ VSI	6.7 ± 0.4	6.0 ± 0.3	6.4 ± 0.1	6.7 ± 0.5	6.8 ± 0.2
**Final Whole Body Composition (% WW)**					
Dry matter	28.0 ± 0.2	28.0 ± 1.8	27.4 ± 0.4	26.9 ± 0.8	27.2 ± 0.7
Crude protein	16.3 ± 0.6	16.0 ± 0.9	15.6 ± 0.5	15.8 ± 0.4	15.7 ± 0.7
Crude fat	8.0 ± 0.7	8.3 ± 1.0	7.9 ± 0.5	7.3 ± 0.3	7.8 ± 0.4
Gross energy (kJ g^−1^)	6.8 ± 0.1	6.8 ± 0.6	6.7 ± 0.2	6.4 ± 0.1	6.6 ± 0.2
Ash	3.6 ± 0.3	3.7 ± 0.2	3.4 ± 0.2	3.6 ± 0.3	3.5 ± 0.2
**Retention per Consumption (% WW)**					
Dry matter	26.8 ± 1.7	26.9 ± 1.6	25.8 ± 0.6	26.1 ± 0.9	25.0 ± 0.3
Protein	36.5 ± 3.3	35.0 ± 1.7	33.3 ± 0.5	35.8 ± 1.0	33.0 ± 1.4
Lipid	107.6 ± 6.7 ^a^	97.1 ± 13.3 ^ab^	85.1 ± 7.5^b^	88.0 ± 3.3 ^ab^	86.3 ± 3.4 ^b^
Energy	29.8 ± 1.2	29.7 ± 2.7	28.1 ± 1.6	28.3 ± 1.0	28.0 ± 0.1
**Gain**					
Dry matter	5.8 ± 0.1	5.7 ± 0.4	5.5 ± 0.1	5.5 ± 0.1	5.5 ± 0.1
Protein (ABW Kg/day)	3.4 ± 0.2	3.3 ± 0.2	3.1 ± 0.1	3.2 ± 0.1	3.2 ± 0.1
Lipid (ABW Kg/day)	1.68 ± 0.15	1.71 ± 0.25	1.58 ± 0.14	1.49 ± 0.05	1.58 ± 0.07
Energy (ABW Kg/day)	1.38 ± 0.01	1.36 ± 0.14	1.31 ± 0.07	1.28 ± 0.01	1.32 ± 0.04
Energy (Total)	6.4 ± 0.2	5.9 ± 0.6	5.5 ± 0.5	5.8 ± 0.2	5.7 ± 0.1

Values are presented as mean ± standard deviation (*n* = 3). Values in the same row without a common superscript letter differ significantly (*p* < 0.05). Absence of superscript indicates no significant difference between treatments ^a^; K: condition factor ^b^; SGR: specific growth ratio ^c^; ABW: average body weight ^d^; FCR: feed conversion ratio ^e^; PER: protein efficiency ratio ^f^; HSI: hepatosomatic index ^g^; VSI: viscerosomatic index.

**Table 7 animals-11-00590-t007:** Instrumental texture and color analysis, water activity (a_W_), and pH of raw fillets of Nile tilapia fed the experimental diets for 10 weeks.

Variable	REF	AOK10	AOK20	CYOK10	CYOK20
**Color Parameters**					
L*	45.7 ± 0.4	45.84 ± 0.4	45.4 ± 0.7	45.6 ± 0.8	45.1 ± 0.6
a*	2.7 ± 0.3	2.9 ± 0.3	2.9 ± 0.3	3.0 ± 0.5	2.8 ± 0.4
b*	−0.8 ± 0.3	−0.3 ± 0.4	−0.5 ± 0.3	−0.6 ± 0.3	−0.3 ± 0.4
C*	2.9 ± 0.6	3.1 ± 1.0	3.0 ± 1.0	3.3 ± 1.0	2.9 ± 0.6
h*	261.5 ± 139.6	272.4 ± 141.0	269.6 ± 142.4	286.3 ± 124.6	213.6 ± 169.8
**Texture Parameters**					
Firmness	58.3 ± 10.0 ^b^	63.7 ± 16.6 ^ab^	65.5 ± 11.0 ^a^	64.8 ± 13.6 ^ab^	61.9 ± 11.0 ^ab^
Cohesiveness	0.42 ± 0.09	0.40 ± 0.05	0.41 ± 0.05	0.41 ± 0.04	0.40 ± 0.03
Elasticity	0.90 ± 0.18	0.93 ± 0.07	0.93 ± 0.06	0.93 ± 0.05	0.93 ± 0.04
Resilience	0.13 ± 0.09	0.14 ± 0.05	0.14 ± 0.05	0.14 ± 0.04	0.13 ± 0.03
**Muscle Analysis**					
a_w_	0.974 ± 0.002	0.978 ± 0.002	0.972 ± 0.003	0.975 ± 0.004	0.976 ± 0.002
pH	6.14 ± 0.07	6.23 ± 0.04	6.25 ± 0.08	6.15 ± 0.06	6.10 ± 0.02

Values are presented as mean ± standard deviation (*n* = 18). Values in the same row without a common superscript letter differ significantly (*p* < 0.05). Absence of superscript indicates no significant difference between treatments. L*: lightness; a*: redness; b*: yellowness; C*: chromaticity; h*: hue angle.

**Table 8 animals-11-00590-t008:** Muscle cellularity of Nile tilapia fed the experimental diets for 10 weeks.

Variable	REF	AOK10	AOK20	CYOK10	CYOK20
Muscle CSA (mm^2^)	512.8 ± 93.0	551.5 ± 64.8	486.5 ± 80.4	534.0 ± 89.2	536.3± 63.0
Total number of fibers × 10^3^ (N)	126.6 ± 29.1	141.0 ± 23.3	123.3 ± 27.6	123.3 ± 26.7	119.5 ± 17.3
Fiber density (N/mm^2^)	244.9 ± 16.4 ^ab^	255.6 ± 27.6 ^a^	252.8 ± 32.1 ^a^	230.4 ± 27.1 ^ab^	223.1 ± 20.9 ^b^
Fiber diameter (µm)	59.6 ± 1.9 ^b^	59.8 ± 3.7 ^b^	60.9 ± 5.4 ^ab^	64.5 ± 3.4 ^a^	63.9 ± 3.0 ^a^
% Fiber ≤ 30 μm	1.7 ± 2.1 ^b^	3.0 ± 1.5 ^a^	4.0 ± 3.4 ^a^	1.7 ± 0.8 ^ab^	0.0 ± 0.0 ^c^
% Fiber > 100 μm	2.5 ± 2.3	1.2 ± 1.1	1.3 ± 1.0	2.3 ± 1.7	2.1 ± 1.9

Values are presented as mean ± standard deviation (*n* = 12). Values in the same row without a common superscript letter differ significantly (*p* < 0.05). Absence of superscript indicates no significant difference between treatments.

## Data Availability

The dataset generated by this study are available from the corresponding author on reasonable request.

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
