# Peer review of "Processed By-Products from Soy Beverage (Okara) as Sustainable Ingredients for Nile Tilapia (O. niloticus) Juveniles: Effects on Nutrient Utilization and Muscle Quality"

_animals, 2021, doi:10.3390/ani11030590_

Round 1

Reviewer 1 Report

The paper evaluates the potential of okara, a soybean by-product, as a protein source in the diets for Nile tilapia. The results show that inclusion of this product up to 20% levels, processed by autoclave or digested with a protease, did not have a negative impact on the growth, feed and nutrient utilization or whole body composition.

The paper is well-written and the results thoroughly discussed. I have only few minor comments.

Methods: Why the authors eventually decided to go for inclusion levels in the growth trial 10 and 20% and not 30% as the digestibility trial?

Line 196: How was the water quality monitored?

Line 213: What was the surface area of the muscle sample?

Figure 1: Please indicate in the figure what was the area the authors measured.

Author Response

Answers for reviewer 1

Point 1: Methods: Why the authors eventually decided to go for inclusion levels in the growth trial 10 and 20% and not 30% as the digestibility trial?

Response 1: This was an author choice for direct comparison with previous manuscripts in the literature including increasing levels of protein sources to replace fish meal. But the use of the protein digestible value of each diet could also be used instead. We considered it would be easier for readers to have these nominal levels, so we prefer to keep as is.

Point 2: Line 196: How was the water quality monitored?

Response 2: The water quality was daily monitored using the parameters of pH, temperature (ºC) and dissolved oxygen (mg/mL). During the growth trial, the values were: pH (6.6 ± 0.1), temperature (24.8 ± 0.4) and dissolved oxygen (7.0 ± 0.5).

Point 3: Line 213: What was the surface area of the muscle sample?

Response 3: For histological evaluation, a one cross-sectional body slab (2 and 3 mm thick) was removed from the region immediately before the first dorsal fin and was photographed together with a scale. Muscle total dorsal cross sectional (CSA) area (mm2) was computed after demarcating the physical limits of the whole dorsal section using the photo taken at sampling time. Muscle CSA of fish fed each diet is reported in Table 8.

Point 4: Figure 1: Please indicate in the figure what was the area the authors measured.

A scale bar is included in Figure 1, so the photo area is not required. Besides, the area of the photo has only be considered to determine fiber density, after counting the number of fibers; muscle CSA was determined as explained above and not in the histological section.

Reviewer 2 Report

Your work is interesting, I noticed few points that I believe I can contribute.
line 19 - remove ", but susceptible to spoilage";
line 307-310 - remove "The proximate composition of the test ingredients is presented in Table 2. The crude protein content of the differently processed okara meals ranged from 19 to 32%, crude fat varied between 9 and 17% and gross energy from 20 to 23 kJ.g-1. ";
line 333-304 - check the average values presented with those presented in the respective table 06 .;
line 427-428 - check the font size;
line 512-524 - the conclusion is very long, there is a lot that is part of the discussion.

Author Response

Answers for reviewer 2

We appreciate the reviewer comments on the manuscript. The changes made in the manuscript are described below in red and were highligted in the manuscript.

Point 1: line 19 - remove ", but susceptible to spoilage";

Response 1: The suggestion was accepted and that part of the sentence was removed from the abstract.

Point 2: line 307-310 - remove "The proximate composition of the test ingredients is presented in Table 2. The crude protein content of the differently processed okara meals ranged from 19 to 32%, crude fat varied between 9 and 17% and gross energy from 20 to 23 kJ.g-1. ";
Response 2: The sentence should be kept in order to further discuss the impact of processing methods on okara nutritional value as highlighted by reviewer 3.

Point 2: line 333-304 - check the average values presented with those presented in the respective table 06 .;

Response 3: The values have been checked and corrected in the text. In fact, a typo was detected and amended. We appreciate the reviewer comment. Line 326: “Final body weight (103.5 ± 31.5 g) and length (16.9 ± 2.3 cm)”

Point 4: Line 427-428 - check the font size;

Response 4: The font size was checked.

Point 5: line 512-524 - the conclusion is very long, there is a lot that is part of the discussion

Response 5: The conclusion has been reduced but keeping major achievements: “The results obtained in this study showed that all different okara meals were well digested by Nile tilapia. The highest protein digestibility was observed in AOK followed by CYOK, demonstrating the effectiveness of the autoclave and the use of a protease from C. Cardunculus without fermentation to increase nutrient bioavailability. The inclusion of up to 20% of AOK or CYOK did not affect fish growth, nutrient utilization, or whole body composition. Fish fed with AOK diets stand out for their high density of muscle fibers, particularly in AOK20 which can explain their high muscle firmness and may result in further hypertrophic growth. Since okara can be obtained from local food industries, its use could decrease the importation of feedstuffs and consequently decrease the carbon footprint in the aquafeed sector.”

Reviewer 3 Report

The authors studied the nutritional potential of a soybean based byproduct, namely okara, as ingredient for Tilapia diets. For this purpose, apparent digestibility (ADCs) of differently processed okara meals were evaluated: dried okara not autoclaved (FOK), dried okara autoclaved (AOK), okara hydrolyzed with Alcalase (ALOK) or Cynara cardunculus protease (CYOK), and hydrolyzed 29 okara fermented with lactic bacteria: Lactobacillus rhamnosus R11 (CYR11OK) or Bifidobacterium ani-30 malis ssp. lactis Bb12 (CYB12OK).

The authors found that processing influenced digestibility of various products. In a following growth inclusion rates of 10-20% of CYOK and AOK did not impact growth of Tilapia or nutrient utilization. Only minor effects on muscle fibre structure and firmness could be detected. The author conclude finally the high potential of presented products for aquafeed sector.

Presented study is of high relevance for the sector and innovative character is given by the products of interests. The study was carefully designed and performed and matches all scientific approaches in general. In addition, the lingual quality and statistical set up are adequate. Neverthelss, there are some points, which should be reflected in the revision process. Therefore, I recommend to accept with minor revisions.

My concerns are as follows:

Introduction

The introduction is a bit too wordy and should be strengthened to the needed informations for explanation of study approach. E.g. line 89 – 105 could be reduced significantly. In addition line 109-113 should be placed in Material and methods section.  

Material and methods   

Line 139-140: how long took fermentation?

Table 2:

It remains unclear, why the nutritional profile in terms of ash, fat, protein of various products differed largely, especially the Cyanara cardunulus products. Why should enzymatic hydrolysis influence ash content, etc.? Please explain! Include in discussion!

How about phytate content? Why not measured?

Table 3:

It is confusing that nutritional composition of presented test diets did not reflect ingredient nutrient profile. E.g. CYB12OK-d has higher protein contents than ALOK-d although CYB12OK shows lower protein contents! Is there any explanation?  

Line 162: Why are the inclusion levels incorporated on nominal levels of 10 or 20%? Why not used on digestible level, as ADC were recorded?

Line172: Nile Tilapia juveniles were monosex all-males?

Line 186: for each test ingredient are new lot of Tilapia were used? In Line 188 is mentioned, that fish fasted between diets? Please clarify!  

Line194. Again … monosex tilapia?

Line212: Why color check? What is ratio behind? I cannot imagine, that okara byproduct could influence flesh colour? Please explain!

Line 287: Explain abbreviations!

Line 296: Add definition of condition factor k!

Results

Table 6: The retention is described on % WW, what does that mean? In addition, the last line in the able is pretty unclear! What does Protein (ABW Kg/day) mean? Please clarify!   

Discussion

See above

Author Response

We appreciate the reviewer comments on the manuscript. The changes made in the manuscript are described below in red and were highligted in the manuscript.

Point 1: Introduction    

The introduction is a bit too wordy and should be strengthened to the needed informations for explanation of study approach. E.g. line 89 – 105 could be reduced significantly. In addition line 109-113 should be placed in Material and methods section.  

Response 1: Line 89-105 this has been reduced. “Nile tilapia, Oreochromis niloticus, is one of the most common finfish species farmed worldwide [31]. It is an omnivorous fish with a dietary protein requirement estimated between 26 and 40% (as fed basis) [32]. Although fishmeal is the best protein source for tilapia and other fish due to its equilibrated amino acid profile, alternative protein sources have been widely considered [10]. Previous studies, showed that the fish meal protein could be replaced by soybean meal in Nile tilapia diets without any adverse effect on growth performance, nutrient digestibility and physiological condition [6, 33,34]. Moreover, the use of soybean by-products is gaining attention due to their large worldwide production and associated environmental problems [35]. Consequently, okara was recently considered a nutritive ingredient for incorporation in fish diets, in particular for omnivorous species like Nile tilapia [36]. The dietary inclusion of okara depends on processing and stabilization techniques able to generate a stable high quality flour [19], but the nutritional value of such feedstuffs for aquafeeds remains to be evaluated.”

Line109-113 These lines have been removed from Introduction section and not placed they were already in Material and methods section.

Point 2: Material and methods   

Line 139-140: how long took fermentation?

Response 2: The fermentations lasted circa 13h. This information was added to the manuscript. Line 134 “…conducted at 37 ºC and 120 rpm, until it reaches a pH of 4.5 (ca. 13 hs).”

Point 3: Table 2:

It remains unclear, why the nutritional profile in terms of ash, fat, protein of various products differed largely, especially the Cyanara cardunulus products. Why should enzymatic hydrolysis influence ash content, etc.? Please explain! Include in discussion!

Response 3: The ash content differs between ingredients due to the different processes they were submitted to. For example, hydrolysed okara has a higher ash content, due to the high content of salts associated to the composition of the buffer solution used during enzymatic hydrolysis. Regarding the fat and protein content, the differences observed are mainly related to the increased level of salts and carbon source incorporated in the fermentation medium, which proportionally reduced fat and protein in dry powder. During the okara fermentation by Lactobacillus rhamnosus R11 and Bifidobacterium animalis ssp. lactis Bb12, a source of sugar (fructose) was added, and this consequently modified the composition of the obtained flour. A line including the % carbohydrates was added to table 2 for better clarity. It is possible to observe that the fermented okara meals (CYB12OK and CYR11OK), showed a higher carbohydrate level than other okara meals, and consequentially lower fat and protein content.  

The values of carbohydrates (%DM) and the calculation equation were added in the table 2.

The Lines 371-377 were included in the discussion: “The present study showed that the proximate composition of okara meal varied according to the technological processing used. The addition of fructose during okara fermentation resulted in higher carbohydrates content in fermented okara meals (CYR11OK and CYB12OK). On the other hand, the use of a buffer solution during the enzymatic hydrolyses resulted in increased ash content in ALOK, CYOK, CYR11OK and CYB12OK meals. These variations led to lower protein and fat contents in fermented okara meals (ALOK, CYOK, CYR11OK and CYB12OK).

Point 4: THow about phytate content? Why not measured?

Response 4: This information was not included in the manuscript as have been previosly described in a first publication (Voss et al., 2018). The antinutrients have been analysed in this previous work and it was possible to observe that due to the heat treatment during the soy beverage production, the values for trypsin inhibitor in okara are lower than the values in raw soybeans. For example, the value for fresh okara was 10.4 ± 0.1 (TUI/mg dried okara), while the autoclaved okara presented 1.0 ± 0.2 (TUI /mg dried okara). Additionally, it was also noted that with the drying process at 65 ºC, the values of TUI decreased to 6.5 ± 0.2 and 0.6 ± 0.2, in fresh and autoclaved okara, respectively. So, the autoclave process practically eliminates okara antinutrients.

In parallel, other authors (Stanojevic, Barac, Pesic, Jankovic, & Biljana V. Vucelic-Radovic, 2013) have also shown that antinutrients in different soybean genotypes presented values between 96 and 198 (TUI/mg  200), while in okara the values were between 4,8 and 8.0 (TUI/mg). Besides, according to Francis, Makkar, & Becker (2001), the thermal treatment used for extraction of soya beverage eliminates most antinutrients, as protease inhibitors and phytates which is an advantage for fish feeding.

Point 5: Table 3:

It is confusing that nutritional composition of presented test diets did not reflect ingredient nutrient profile. E.g. CYB12OK-d has higher protein contents than ALOK-d although CYB12OK shows lower protein contents! Is there any explanation?  

Response 5: The formulas were well designed and followed two completely different approaches. It is clear that we have ingredients with different nutritional values. Their digestibility was determined using a traditional approach in fish, that is by replacing 30% of the whole formulated mixture by each ingredient. Consequently, the nutritional value of each diet used in the digestibly study varied. However, when those ingredients are considered in a dietary formulation for a growth trial, diets can only be compared if they are nutritionally equivalent in terms of protein and energy. So, diets were formulated using distinct ingredients, but keeping all diets isonitrogenous. As can be observed in Table 4, the addition of okara was mainly achieved at the expense of different levels of soybean meal taking into consideration the composition of each ingredient.

Point 6: Line 162: Why are the inclusion levels incorporated on nominal levels of 10 or 20%? Why not used on digestible level, as ADC were recorded?

Response 6:This was an author choice for direct comparison with previous manuscripts in the literature including increasing levels of protein sources to replace fish meal. But the use of the protein digestible value of each diet could also be used instead. We considered it would be easier for readers to have these nominal levels, so we prefer to keep as is.

Point 7: Line172: Nile Tilapia juveniles were monosex all-males?

Response 7:This information has been added to text: Line 166: “....with Nile tilapia (O. niloticus) juveniles (monosex all-male population) produced in the facility …”

Point 8: Line 186: for each test ingredient are new lot of Tilapia were used? In Line 188 is mentioned, that fish fasted between diets? Please clarify!

Response 8: The digestibility trial followed very well established methodologic approaches previously described elsewhere (Pereira et al., 2012). As the recirculating water system used was only constituted by three tanks, this procedure had to be divided into four periods of faeces collection, for replication of results (n=3). In each period, a new lot of fish, with the same size-range, from the same monosex all-male initial population was selected, and fed a new experimental diet, until obtaining three replicates for each test diet. Each replicate was carried out in a different tank to reduce any tank effect. A 24h fasting is recommended after manipulation of the fish (in this case change of tank) and prior distribution of a new experimental diet. We have rephrased the text for clarity.

Point 9: Line 212: Why color check? What is ratio behind? I cannot imagine, that okara byproduct could influence flesh colour? Please explain!

Response 9: The colour parameter is considered an important sensory parameter and can be affected by dietary ingredients that may affect flesh quality. The colour depends on pigments, but also on fat deposition in the muscle and this can be largely affected by vegetable ingredients with variable composition.

Point 10: Line 287: Explain abbreviations!

Response 10: It has been added.

Line 277: “(Dry matter (DM) basis)”

Line 279: “Nitrogen (N), Protein (P), Lipid (L) or Energy (E) gain…”

Point 11: Line 296: Add definition of condition factor k!

Response 11: Fulton's condition factor, K, was proposed by Fulton in 1904, and it assumes that the standard weight of a fish is proportional to the cube of its length. K is a measure of an individual fish's health status. This indicator is largely known so we believe that the definition is not required in the manuscript.

Point 12: Results

Table 6: The retention is described on % WW, what does that mean? In addition, the last line in the able is pretty unclear! What does Protein (ABW Kg/day) mean? Please clarify!   

Response 12:The calculation of nutrient balance is widely used in nutritional studies as can be confirmed in the literature (Batista et al., 2020; Campos, Matos, Marques, & Valente, 2017). DM, L, P or E retention was calculated as follows:  100 x (final body weight × final carcass nutrient content) – (initial body weight × initial carcass nutrient content) /DM, lipid, protein or energy intake; it is expressed as the percentage of each nutrient that was retained by fish in terms of wet weight (% WW). As explained in the M&M section, Average body weight (ABW) = (final body weight + initial body weight)/2; In table 6 information was now added to the footer to include all abbreviations used in the table. Besides, we have realized that a line was missing related to nutrient gain. This was now amended.